# Black Soldier Fly, *Hermetia illucens* as an Alternative to Fishmeal Protein and Fish Oil: Impact on Growth, Immune Response, Mucosal Barrier Status, and Flesh Quality of Juvenile Barramundi, *Lates calcarifer* (Bloch, 1790)

**DOI:** 10.3390/biology10060505

**Published:** 2021-06-07

**Authors:** Amanda Hender, Muhammad A.B. Siddik, Janet Howieson, Ravi Fotedar

**Affiliations:** 1School of Molecular and Life Sciences, Curtin University, 1 Turner Avenue, Bentley, WA 6102, Australia; 2Department of Fisheries Biology & Genetics, Patuakhali Science and Technology University, Patuakhali 8602, Bangladesh

**Keywords:** Insect larvae, fishmeal replacement, intestinal morphology, acidic mucin, fatty acid, cytokine expression, *Lates calcarifer*

## Abstract

**Simple Summary:**

Fishmeal and fish oil being recognized as nutritionally balanced and preferred protein and lipid sources in aquafeeds for finfish farming. However, the limited supply, soaring prices, and depletion of wild capture fisheries have prompted the aquaculture industry to look for widely available and less expensive alternatives for fishmeal and fish oil to ensure the sustainability of aquaculture production. As an alternative to fishmeal protein and oil, black soldier fly (BSF) larvae is considered a promising candidate for its high protein, lipid and minerals content, as well as for its bioactive potential with anti-microbial, anti-fungal, and anti-viral functions. This study investigated the effects of partial substitution of fishmeal protein and fish oil with partially defatted BSF larvae protein and oil on growth performance, non-specific immunity, mucosal barrier status and muscle fatty acids composition in barramundi. The findings of the study confirmed that BSF could substitute 30% fishmeal and fish oil successfully without compromising the growth of barramundi. Further, partial replacement of fishmeal and fish oil with BSF diets improved bactericidal activity, immune-related cytokine expression, and mucin cells in both gut and skin which are an indication of improved immunity of barramundi.

**Abstract:**

A feeding trial was conducted to test the effects of partial replacement of fishmeal (FM) protein and fish oil (FO) with partially defatted black soldier fly, *Hermetia illucens* insect protein, and oil, respectively, on growth performance, immune response, gut and skin barrier status, and flesh quality in juvenile barramundi. Four isonitrogenous and isocaloric diets used in the study were a control diet based on FM, 30% FM replaced with *H. illucens* protein (HiP), 30% FO replaced with *H. illucens* oil (HiO), and both 30% FM and 30% FO replaced with *H. illucens* protein and oil (HiPO). Diets were fed twice a day to satiety in triplicated groups of barramundi with an initial body weight of 1.74 ± 0.15 g per fish. At the end of the trial, growth and feed utilization indices were found insignificant (*p* > 0.05) between the test diets and control. A significant increase in bactericidal activity was observed in fish fed the HiP diet while serum lysozyme activity was unchanged. Stress-related heat shock proteins (HSP70 and HSP90) did not differ significantly among the test diets while immune-relevant genes (IL-1β and IL-10) were significantly upregulated in HiP and HiOP groups. The number of mucin cells were increased in the gut and skin of HiP and HiOP fed fish when compared to the control diet. The total fatty acid compositions (∑SFA, ∑MUFA, ∑PUFA, ∑n-3, and ∑n-6) in the muscles of barramundi were not significantly influenced with *H. illucens* protein and oil diets when compared to the control.

## 1. Introduction

Fishmeal (FM) and fish oil (FO) are recognized as ideal sources of crude protein and crude fat in aquafeeds [1], but the scarcity of raw materials compelled aquaculture scientists to search for available ingredients as sustainable source of FM and FO in formulating aquafeeds [1]. In the past, many researchers have examined the efficacy of various plant protein sources in aquafeeds but less information has been published regarding the inclusion of both animal protein and oil from the same animal source like wild fish in aquafeeds. In recent years, a variety of low-cost animal by-products-based diets incorporating ingredients such as insects meal, poultry by-product meal, meat and bone meal, and fish offal meal have received attention as possible alternative protein sources in fish diets [2,3]. In particular, insect larvae are considered a promising, good quality, efficient, and sustainable alternative protein ingredient that is yet to be exploited to its full potential [4]. Insect larvae may contain high protein, fats, and minerals, making them an attractive substitute to FM in aquafeeds [5]. However, the nutritional composition of insects depends on the selection of insects that are similar to FM in terms of their essential amino acids (EAA), phospholipids, and fatty acids components [5]. Hence, black soldier fly (BSF), *Hermetia illucens*, belonging to the insects taxonomic order Diptera has been suggested as a promising alternative for the aquaculture industry since it contains 42% crude protein and 35% fat, and shows a very similar pattern of EAA to FM [6,7]. A number of past studies have revealed the suitability of BSF larvae as an ingredient in the nutrition of rainbow trout, *Oncorhynchus mykiss* [8,9], juvenile turbot, *Psetta maxima* [10], Pacific white shrimp, *Litopenaeus vannamei* [11], juvenile jian carp, *Cyprinus carpio* var. Jian [6,12], European seabass, *Dicentrarchus labrax* [13], clownfish, *Amphiprion ocellaris* [14], and barramundi, *Lates calcarifer* [15].

Although beneficial effects have been observed in some fish species utilizing BSF larvae as an alternative to FM [7,16], some potential impediments of using BSF larvae in aquaculture are palatability, digestibility, chitin content, bioaccumulation of toxicity, and deficiencies in EAA or long-chain fatty acids [7]. Furthermore, the presence of high percentages of lipids in the BSF larvae may incur milling difficulties in the feed industry, for example, susceptibility to oxidation, excessive energy, and a decrease in pellet stability [7]. These hindrances can, therefore, limit the utilization of BSF larvae, as substitution of FM, in aqua-feeds. However, some processing techniques like hydrolysing, drying, defatting, or ensiling could be used to improve palatability, digestibility, and nutrient availability [17]. Among these techniques, the defatting process could be an ideal solution from a fish nutrition perspective as it can lead to protein level over 60% [18] and results in a well-balanced essential AA and FA composition [4], particularly for marine fish [7]. Additionally, BSF larvae oil, a potential by-product of BSF larvae meal production may contain a high percentage of fat which is rich in medium-chain fatty acids (MCFA) with lauric acid (LA, 12:0) reported to be between 21.4–49.3% of the total fatty acids [6]. Therefore, more applied research is needed to investigate the potential to increase the inclusion levels of BSF larvae oil in aquadiets. However, the composition of BSF larvae oil may vary depending on the medium in which the BSF larvae are being cultured [19]. For instance, BSF-fed fish offal can increase the mega-3 long chain-polyunsaturated fatty acids (LC-PUFA) in larvae [19] and BSF larvae cultured in the seaweed-based medium can enrich larvae with eicosapentaenoic acid (EPA) and iodine [20].

*Lates calcarifer*, barramundi is a high-profit aquaculture fish that is farmed across Australia and the Indo-Pacific regions. According to the Australia Barramundi Farmers Association [21], barramundi production in Australia is projected to fetch an economic value exceeding US$250 million by 2025. To maintain the sustainability of barramundi production, studies have been conducted over the years to investigate the efficient use of insect meal over expensive FM protein sources [22]. To date all studies in barramundi fed insect meal utilized BSF raw product and emphasis was only given to the growth performance and feed utilization of the fish [15]. Studies on the utilization of defatted insect meal and the aligned oil fraction in the diet of juvenile barramundi are limited. Therefore, the present study was designated to investigate the effects of replacing both FM protein and oil with defatted BSF larvae meal protein and oil on immune response, gut and skin mucosal status, and muscle fatty acids composition along with the growth performance of barramundi.

## 2. Materials and Methods

### 2.1. Experimental Diets

Four different isonitrogenous and isolipidic diets were formulated and prepared with approximately 47.0% crude protein and 13.0% crude lipid in the diets. The *H. illucens* larvae were supplied by Future Green Solutions as 6-day larvae cultured on a 50% carp-based substrate. The larvae were dried by Specialty Feeds, 3150 Great Eastern Hwy, Glen Forrest. Western Australia, using an industrial drier and then defatted using an oil extractor press that resulted in approximately 80% extraction levels. Four isonitrogenous and isocaloric diets were formulated including a FM based diet that served as control, 30% FM protein replaced with partially defatted *H. illucens* protein (HiP), 30% FO replaced with *H. illucens* oil (HiO), and 30% FM and 30% FO together replaced with *H. illucens* protein and oil (HiPO). Experimental diets used in feeding the fish throughout the experiment period were pelletized to 2.5 mm. All diets stored at −20 °C until use in trial. The experimental design and condition with detail about dietary codes for feeds are displayed in Figure 1. Diet formulation, all dietary ingredient and nutritional parameters are presented in Table 1. The amino acid and fatty acid compositions of full fat and defatted black soldier fry, and FM are presented in Table 2.

### 2.2. Animal Husbandry and Feeding

Barramundi fry were obtained from the Mainstream Aquaculture Group Pty Ltd., Werribee in VIC, Australia and transported in oxygenated plastic bags filled with 1/3 of water at 5 ppt salinity. The fish were transported to CARL, Curtin University, Australia, where fish were transferred into 250 L tank for two weeks acclimation. During the acclimation period, fish were fed a commercial barramundi diet (Skretting, Australia). After the acclimation, the fish were then distributed into the 12 experimental tanks in recirculatory aquaculture system (RAS) at a stocking density of 20 fish per tank with an average individual weight of 1.74 ± 0.15 g per fish. Experimental diets were fed randomly to triplicate groups of fish to satiation and feeding was done twice daily, in the morning at 9:00 am and evening 5:00 pm throughout the experimental period of 42 days. All experimental units were continuously aerated with air stones placed at the bottom of the tank to maintain dissolved oxygen (DO) levels at optimum and filtration of the culture tanks was carried out using standard mesh biological sponge filters (XY-380). Water temperature in experimental units and 24-h room light and dark condition (14 h dark and 10 h light) were also uniformly maintained throughout the experiment period with the aid of electric heaters and fluorescent light. A water exchange of 50% was completed every 3 days while the DO, temperature, ammonia (NH_3_), nitrite (NO_2_^−^), and nitrate (NO_3_^−^) concentrations were measured and recorded daily. The measured water quality parameters were DO 6.42 ± 0.24 mg L^−1^, temperature 27.15 ± 1.72 °C, salinity 33.10 ± 2.49 ppt, ammonia nitrogen 0.42 ± 0.26 mg L^−1^, and nitrite 0.36 ± 0.45 mg L^−1^. These results were within the suitable range of fish culture [23,24]. DO and temperature were measured with an Oxyguard Polaris 2 D.O. Meter and the NH_3_, NO_2_^−^ and NO_3_^−^ levels were measured using API liquid test kits.

### 2.3. Blood Extraction and Serum Biochemistry

Five juvenile barramundi from each tank were placed into an anaesthetic bath of AQUI-S at a sedative dose of 10 mg/L until loss of equilibrium was noted, fish were placed on a moist towel and blood was extracted into an Eppendorf tube by puncturing the caudal vein peduncle using a 1-mL plastic syringe and a 22G×1/2′′ straight needle. The blood was left at room temperature for 24 h to clot and then tubes were placed in a centrifuge for 15 min at 3000 rpm at 4 °C. The serum was then separated and stored in a −80 °C freezer for further analysis. Serum biochemical parameters including aspartate aminotransferase (AST), glutamate dehydrogenase (GLDH), cholesterol, triglyceride (TG), total protein, and albumin were determined using an automated blood analyser (SLIM; SEAC Inc., Florence, Italy) in accordance with the protocols from [25]. The albumin and globulin ratio (A/G ratio) were calculated by dividing the total albumin content by the difference of total serum and albumin protein values. Serum lysozyme and bactericidal activities were performed following the previously described protocol of Siddik et al. [26].

### 2.4. RNA Extraction and qRT-PCR Analysis

Liver and intestine were collected from six euthanized (AQUI-S, 175 mg L^−1^) fish that were used for blood collection and stored in RNA Later (Sigma-Aldrich, Germany) at −80 °C until RNA extraction. Samples were homogenised immediately in tissue lyser with sterile beads (Qiagen, Hilden, Germany) and 5 mg tissue from each sample was used for RNA extraction following manufacturer’s instructions of RNeasy Mini Kit (Qiagen, Hilden, Germany). RNA integrity and quality were checked in agarose gel and NanoDrop spectrophotometer 2000 c (Thermo Fisher Scientific, Waltham, MA, USA). Complementary DNA (cDNA) was synthesized from 1 µg of RNA as per the manufacturer’s protocol of Omnicript RT kit (Qiagen, Hilden, Germany).

Quantitative real-time PCR (qRT-PCR) for HSP70, HSP90, IL-1β, IL-8, IL-10, and TNF-α (Table 3) was performed using cDNA samples of the liver and distal intestine in 7500 Real-Time PCR System (Applied Biosystems, Foster City, CA, USA) employing PowerUp^TM^ SYBR Green Master Mix (Thermo Scientific, Waltham, MA, USA) as per the manufacturer’s protocols. The total volume of PCR reaction was 20 µL consisting of 10 μL TransStart Top Green qPCR SuperMix (2×), 0.6 μL (5 μM) of each primer, 1 μL cDNA, and 7.8 μL RNase-free water. The quantitative real-time PCR data were analysed with 2^−^^△△^^CT^ method using β-actin as a reference gene.

### 2.5. Gut and Skin Mucosal Response

At the end of the trial, six randomly selected individual fish per treatment, two from each replicate, were used for histological analysis. The skin and distal gut samples were processed following the standard histological procedures. The dissected skin and gut fragments were first preserved in 10% buffered formalin and then dehydrated by graded ethanol and cleared by xylene and embedded in paraffin wax for histological examination. The fixed tissues were then cut into 5 µm in size by a rotary microtome and stained with Periodic Acid-Schiff (PAS). The histological slides were photographed under a light imaging microscope (BX40F4, Olympus, Tokyo, Japan). The gut and skin mucosal measurement and number of mucin cells were counted following the protocol of our earlier study [22].

### 2.6. Fatty Acid Composition

At the end of the trial, four fish from each tank were euthanized and filleted. Fillets were freeze dried for 3 days at −48.4 °C and 1.9 × 10^−1^ mB, pooled together, ground into a fine powder, and stored at −80 °C for further analysis. Pooled muscle fatty acids composition was analysed using methylated ester method following the procedures described by [31].

### 2.7. Calculations

Growth performance in terms of weight gain, specific growth rate, feed conversion ratio, and survival were calculated as follow:Weight gain (WG, g)=[(Mean final weight−Mean initial weight)/(Mean initial weight)]
Specific growth rate (SGR, %⁄d)=[(ln (final body weight)−ln (pooled initial weight))/Days]×100
Feed conversion ratio (FCR)=[(Dry feed fed)/(Wet weight gain)]
Survival (SR, %)=[(Final number of fish)/(Initial number of fish)]×100

The fillet lipid quality of barramundi fed experimental diets were determined using two important lipid indexes, atherogenicity (AI) and thrombogenicity (TI), as follows:AI=(aC12:0+bC14:0+C16:0)/(dp+eM+FM′)
where, P is the sum of n3 and n6 PUFA; M is the oleic acid and M’; is the sum of other monounsaturated fatty acids (MUFA); a, b, C, d, e, and F are empirical constants; b = 4 and other constants = 1.
TI=(C14:0+C16:0+C18:0)/[(nM+nM′+p(n6)+q (n3)+(n3/6)]
where M and M’ are as before; n, o, p, and q are empirical constants; n, o, p = 0.5 and q = 3.

### 2.8. Statistical Analysis

Unless specified otherwise, all analysis results are presented as mean ± SE. All data are subjected to normality and homogeneity of variances with Shapiro–Wilk’s and Levene’s tests. When both tests were satisfied, a one-way ANOVA followed by Tukey multiple range test was performed on growth performance, blood biometry indices fatty acid composition, and immune responses to test the significant differences between the experimental groups at *p* < 0.05.

## 3. Results

### 3.1. Growth Performance and Feed Utilization

The dietary inclusion of defatted BSF larvae meal and oil did not influence the growth performance in terms of FBW, WG, and SGR. Mean FCR ranged between 0.76 and 0.98 with no significant difference among the test diets (Table 4). Survival rate of fish in culture units during the entire experimental duration was also not significantly affected by the different dietary treatments.

### 3.2. Blood and Serum Biochemical Response

Blood and serum biochemical indices including glucose, AST, GLDH, cholesterol, triglyceride, creatinine, glucose, total protein, albumin, globulin, and albumin/globulin ratio (A/G) were not influenced by the FM and FO replacement diets with BSF larvae with the exception of urea which increased significantly in barramundi fed HiO and HiPO diets (Table 5). Serum lysozyme activity showed no variations amongst dietary treatments whilst bactericidal activity was significantly higher in HiP than the control (Figure 2).

### 3.3. Hear Shock Protein and Cytokine Gene Expressions

Although heat shock proteins (HSP70 and HSP90) in the liver were unaffected by test diets, the immune-relevant cytokines (IL-1β and IL-10) were significantly upregulated in the distal intestine of barramundi (Figure 3).

### 3.4. Gut and Skin Mucosal Morphology

The histochemistry of gut and skin in fish fed control and BSF diets, and their statistics are shown in Figure 4. The number of mucin cells in the gut and skin of fish fed HiP and HiPO diets increased significantly while gut and skin muscular layer thickness, villus height and width were not affected by test diets when compared to control.

### 3.5. Muscle Fatty Acids Composition

The results of fatty acids compositions in barramundi muscle from the different diets is presented in Table 6 and Figure 5a. Total fatty acids content including saturated fatty acids (SFA), polyunsaturated fatty acids (PUFA), monounsaturated fatty acids (MUFA) were not influenced by BSF dietary inclusion (Figure 5a). n3/n6 ratio increased significantly in the muscle of barramundi fed HiP diets but was not increased in HiO and HiPO fed groups compared to the control (Figure 5b). Lipid quality indices including atherogenicity and thrombogenicity elevated significantly in HiP and HiPO fed groups compared to the control (Figure 5c,d).

## 4. Discussion

A major concern in the aquaculture sector is the dependency on FM and FO, and a desirable goal is to search for more sustainable, cost effective, and environmentally friendly ingredients. Searching for these alternatives is challenging as alternatives must provide required levels of essential amino acids, PUFAs, and high nutrient and energy bioavailability to ensure adherence to the fish health and welfare standards [32]. In comparison with other conventional feed commodities, insects have been receiving attention as candidate feed ingredients due to their minimal impact on the environment [33,34]. In particular BSF larvae meal can meet the nutritional requirement of many terrestrial and aquatic animals, including fish [35]. Katya et al. [15] evaluated BSF larvae meal in the diet of juvenile barramundi and reported that up to 28.4% can be replaced without compromising growth performance. In the present study, diets containing either 30% BSF protein or oil showed no significant difference than the FM based control diet. Similarly, dietary inclusion of partially defatted BSF larvae meal in the diet of Japanese seabass *Lateolabrax japonicas* (up to 64%) [36]. European sea bass, *Dicentrarchus labrax* (up to 64%) [37] and rainbow trout *Oncorhynchus mykiss* Walbaum (up to 50%) [38] did not influence the growth performance. However, feeding African catfish, *Clarias gariepinus* with 50% partially defatted BSF larvae meal increased the growth performance significantly [39] while inclusion of 30% partially defatted BSF larvae meal impaired the growth performance of juvenile meagre, *Argyrosomus regius* when compared with the control [40]. These mixed results may be due to different fish species and different culture methods or feeding substrate used for BSF larvae production.

Serum biochemical parameters have been used as an important indicator to assess the fish health status when alternatives to FM ingredients are used in aquafeed. Increased levels of AST and GLDH in fish blood serum are highly associated with liver damage or necrosis. In the present study, dietary inclusion of BSF larvae meal and/or oil did not affect the AST and GLDH level in the blood serum, indicating that liver health was not affected by the test diets. Similarly, dietary intake of defatted BSF larvae meal ranging from 25 to 100% did not increase the AST level in the serum of Jian carp, *Cyprinus carpio* [6]. Presence of chitin in BSF larvae has been reported to have triglyceride and cholesterol lowering capacity, such effects were not observed in the present study, similar to the following studies [6,33,41]. Blood urea nitrogen and creatinine, nitrogenous end products of metabolism, provide a very accurate estimation of the kidneys function [42]. Although urea was significantly affected by the inclusion of 30% BSF oil, none of the test diets had significant effect on creatinine levels. This result suggested that 30% oil from BSF could hamper the kidney function which deserves further investigations to optimise the inclusion level of BSF oil in the diet of barramundi. In agreement with the reports [6,9,37], dietary inclusion of defatted BSF protein and/or in combination with BSF oil did not affect the serum total protein, albumin, globulin, and A/G ratio of barramundi. The observed serum biochemical indices herein were within the normal range determined for barramundi [43].

Insects contain silkose or dipterose exerting immune-stimulating activity in mammals [44]. The inclusion of low levels of chitin, derived from crustacean, elevated the immune response in fish [45], and similar effects could be expected from insect chitin [35]. However, very few results are available on the effects of BSF larvae meal on the immune response of aquatic animals [36,46]. Wang et al. [36] reported that addition of graded levels of defatted BSF larvae meal (from 16% to 64%) did not enhance the lysozyme activity in the serum of juvenile Japanese seabass, *Lateolabrax japonicas,* similar to the present study. However, Chaklader et al. [22] reported higher levels of lysozyme activity in barramundi when diets containing 45% poultry by product meal was supplemented with 10% full fat BSF, indicating that supplementation, rather than protein replacement with BSF larvae meal, could boost the lysozyme levels. Many researchers have recommended the incorporation of BSF larvae at lower levels to promote the growth and health of fish and shellfish [22,46,47]. It has been suggested that BSF larvae meal could contain novel antimicrobial peptides, characterised by an inhibitory response to Gram-positive and Gram-negative bacterium, and fungus [40,48]. In the present study, the bactericidal activity increased significantly in fish fed HiP. Similarly, supplementation of 10% BSF larvae with poultry by-product meal modulated bactericidal activity in the serum of barramundi, *L. calcarifer* [22]. Increased and rapid antibacterial activity was observed in the serum of rainbow trout, *Oncorhynchus mykiss* when fed 25% and 50% of yellow mealworm, *Tenebrio molitor* larvae meal [7]. The insignificant effects of HiO alone or combined with HiP on lysozyme and bactericidal activity is similar to the findings of [6] who reported that up to 100% of BSF larvae oil did not affect the immune response of juvenile Jian carp, *Cyprinus carpio* var. Jian.

Substitution of FM with alternative protein ingredient have been reported to induce the expression of hepatic stress-related gene such as HSP70 and HSP90 in barramundi [22]. HSP70 and HSP90 are two important heat shock proteins, showing a lower level of expression under normal conditions but is induced rapidly to various stressor and suboptimal conditions of diet [49,50]. For instance, expression of HSP70 increased gradually with increasing levels of partially defatted BSF larvae meal (25 to 100%) in Zebrafish [51] and its expression also increased in the liver of Jian carp, *Cyprinus carpio* var. Jian when fed increasing levels of partially defatted BSF larvae meal (50–100%) [6]. However, Chaklader et al. [22] reported that barramundi fed 45% poultry by-product meal diet with the addition of 10% BSF larvae protein did not changes the expression of HSP70 and HSP90 when compared to FM-based control. In the present study, barramundi fed 30% HiP and/or HiO showed a similar expression of HSP70 and HSP90 in liver, suggesting that 30% protein and lipid from FM and FO could be replaced with HiP and HiO without imposing any stress. Unlike the heat shock proteins, immune relevant genes (IL-1β and TNF-α) were showed a significant difference among the test diets, similar to the findings of [51]. Another study was conducted by the same authors changing substrate of BSF larvae culture and found that dietary inclusion of BSF larvae meal reared on roasted coffee by-product and *Schizochytrium* sp significantly increased the expression of immune-relevant genes including IL-1β, IL-8, TNF-α, and IL-10 in the intestine of Zebrafish [51]. Here we found upregulated expression of IL-1β and IL-10 in the intestine of fish fed BSF diets. All insects-based diets contain chitin [52], a sensitive molecule in aquafeeds which has positive effects on fish immune system [7,51], while chitin at higher inclusion levels (>30%) may induce intestinal inflammation and a decline in nutrient assimilation [10,13,53]. In the present study, the lower inclusion of insect meal resulting in lower levels of chitin showed no negative effects on the stress-relevant and immune genes in barramundi. However, a long-term study is needed covering the whole life cycle of barramundi, from larvae to adult, fed increasing levels of BSF larvae meal to provide more information on heat shock protein and immune-relevant gene expression.

The histological analysis of intestinal trophism has been considered a good indicator to evaluate the nutritional status of fish [54,55,56]. A number of previous studies have demonstrated that a proper orientation of intestinal cells/the intestinal mucosal morphology is associated with the higher nutrient utilization and absorption leading to improved health and immune functions. In the present study, the number of mucin cells were significantly increased in the distal intestine of barramundi fed HiP and HiPO diets. This result is consistent with a previous study that found that the dietary supplemented of BSF in PBM diets did improve the number of goblet cells in barramundi [57]. A recent study of [22] reported that supplementation of BSF in barramundi diet improved the skin mucosal barrier functions in barramundi, manifested by increasing number of acidic mucins which may produce mucus facilitating many biological activities to prevent the colonization of infectious microorganisms. The stimulated skin mucosal barrier function could be due to the presence of antimicrobial peptides and other polysaccharides in HI larvae [58]. However, further study is needed to provide more information how these components in BSF larvae stimulated the skin barrier function in fish.

Fatty acid composition of fish muscle has been reported to be influenced by dietary inclusion of BSF larvae meal in aquadiets [14,38,51]. Dietary inclusion of BSF larvae meal resulted in no significant effect on the total SFA content; however, the lauric acid (C12:0) level was augmented in the muscle of fish fed BSF larvae meal and oil. Such changes may be due to the presence of higher level of C12:0 in BSF larvae, which were reported by many earlier studies [14,20,59]. The incorporation of BSF larvae in the diet of fish increased the total SFA content in muscle with a concomitant decrease in MUFA and PUFA contents [12,38,59]. However, in the present study, replacement of FM with BSF larvae meal imposed no effect on MUFA and PUFA contents. Similarly, MUFA and ∑n-6PUFA in the whole-body of juveniles meagre, *Argyrosomus regius* were unaffected by dietary inclusion of 10–30% partially defatted BSF meal [40]. Indices including AI and TI calculated based on SFA, MUFA, and PUFA are highly correlated with human nutrition. Although AI and TI of the flesh from the barramundi fed HiP and HiO diets differed significantly, the values achieved were all less than 1, indicating no harmful effects for human consumption [38]. Further research should investigate the role of chitin and bioactive polysaccharides in BSF larvae in fish health.

## 5. Conclusions

The substitution of 30% FM protein and FO with partially defatted BSF protein and extracted oil is possible in the diets of barramundi with no deleterious effects on the biological performance and flesh quality. Serum lysozyme activity is unaffected while bactericidal activity increases in 30% BSF protein fed barramundi. Gut and skin mucosal immunity as determined by estimating the number of acidic mucins increased significantly in response to BSF supplemented diets. The expression of certain immune related cytokine genes is significantly higher in fish fed insect-based diets. The lipid indices are unchanged in the muscle of barramundi fed insect-based diets indicate that there is no harmful effect on humans if BSF fed barramundi are consumed.

## Figures and Tables

**Figure 1 biology-10-00505-f001:**
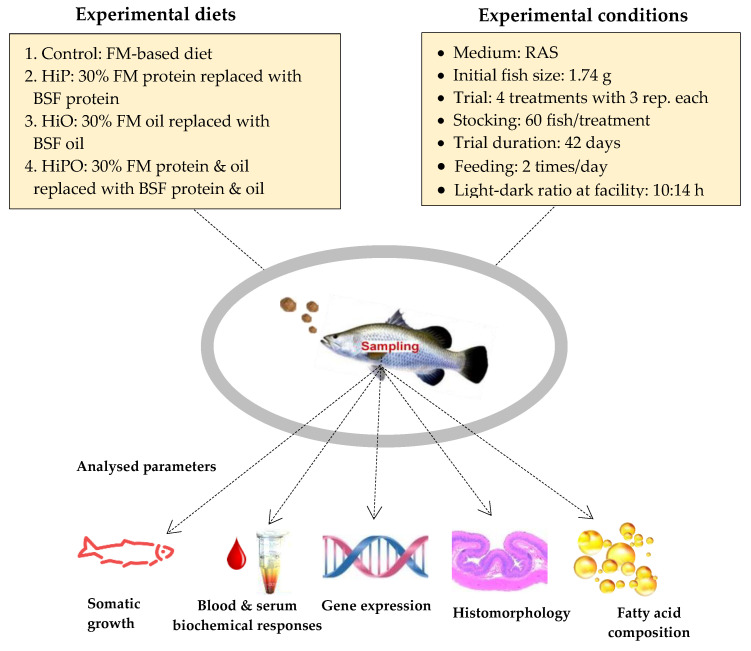
Experimental design.

**Figure 2 biology-10-00505-f002:**
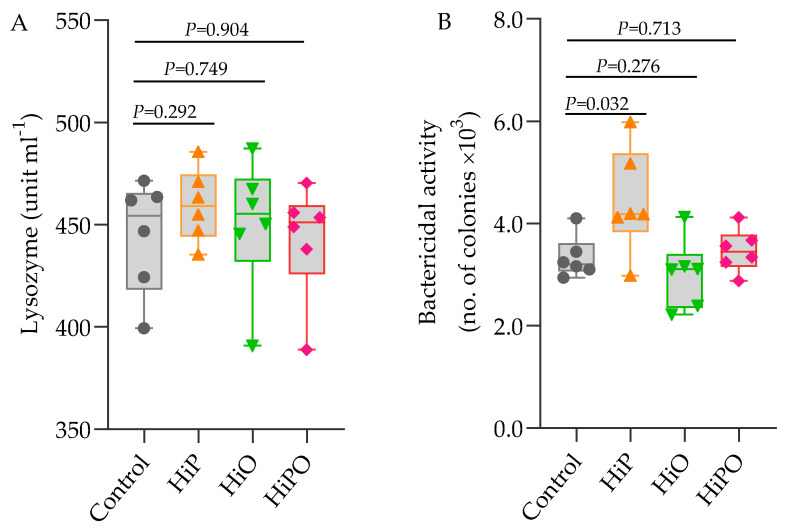
Serum lysozyme and bactericidal activity of barramundi fed BSF larvae protein and oil as FM protein and oil replacement diet. Data were expressed as mean ± SE, *n* = 6. Bar holding *p*-value denotes significant level among the experimental treatments (one-way ANOVA; Tukey post-hoc test; significant at *p* > 0.05).

**Figure 3 biology-10-00505-f003:**
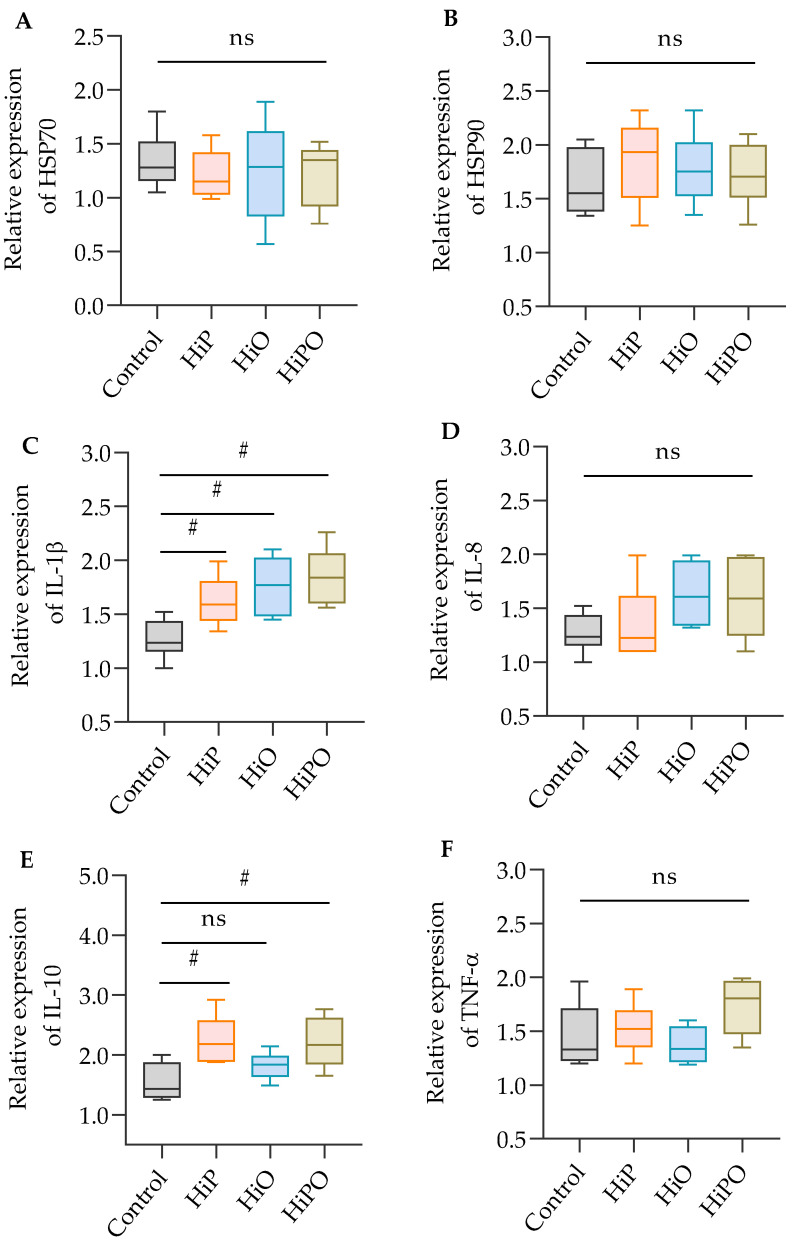
Relative expression of heat shock proteins, HSP70 and HSP90 (**A**,**B**) in the liver and immune-responsive genes IL-1β, IL-8, IL-10, and TNF-α (**C**–**F**) in the distal intestine of juvenile barramundi fed BSF larvae protein and oil as FM protein and oil replacement diets. Data were expressed as mean ± SE, *n* = 6. Bar indicates significant difference among the experimental treatments (one-way ANOVA; Tukey post-hoc test; ns, not significant at *p* > 0.05; #, significant at *p* < 0.05).

**Figure 4 biology-10-00505-f004:**
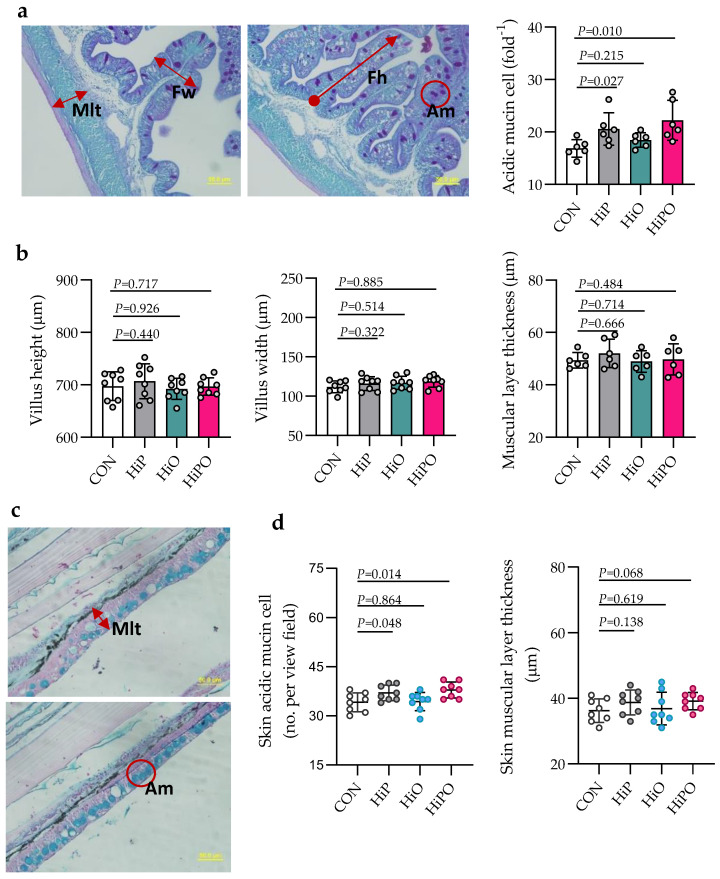
Gut and skin mucosal morphology of barramundi fed partially defatted BSF larvae protein and oil as FM protein and oil replacement diets. (**a**) Schematic representation of histometric measurement of muscular layer thickness (Mlt), fold height (Fh), fold width (Fw), and acidic mucins (Am) of distal gut (PAS, 40× magnification, scale bar 50 µm). (**b**) Quantification of Mlt, Fh, Fw, and Sam of the distal gut of barramundi. (**c**,**d**) Skin histometric measurement and quantification of skin muscular layer thickness (Mlt) and number of skin acidic mucin (Am) (Alcian Blue pH 2.5 stain, 40× magnification). Data were expressed as mean ± SE, *n* = 6. Bar with *p*-value denotes significant level among the experimental treatments (one-way ANOVA; Tukey post-hoc test; significant at *p* > 0.05).

**Figure 5 biology-10-00505-f005:**
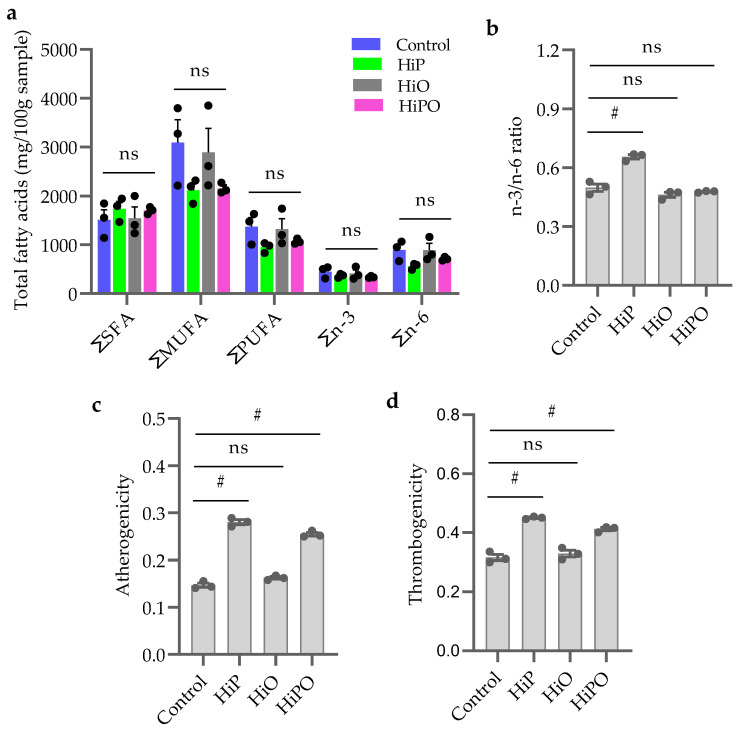
Total fatty acids composition (**a**,**b**) and lipid indexes (**c**,**d**) in the muscle of barramundi fed partially defatted BSF larvae protein and oil as FM protein and oil replacement diet. Data were expressed as mean ± SE, *n* = 3. Bar holding different indications denote significant level among the experimental treatments (one-way ANOVA; Tukey post-hoc test; ns, not significant at *p* > 0.05; #, significant at *p* < 0.05).

**Table 1 biology-10-00505-t001:** Ingredients and nutritional composition of experimental diets fed to juvenile barramundi, *Lates calcarifer*.

Ingredients (g Kg^−1^ of DM)	Control	HiP	HiO	HiPO
FM ^†^	495.00	346.50	497.60	346.50
Partially defatted BSF larvae ^‡^	0.00	221.00	0.00	219.00
Wheat flour	126.20	65.10	121.00	65.50
Wheat starch	82.00	70.00	84.60	72.20
Soybean meal	103.00	103.00	103.00	103.00
Wheat gluten	150.0	150.0	150.0	150.0
Fish oil	37.90	38.50	27.40	27.40
BSF larvae oil	0.00	0.00	10.50	10.50
Trout vitamin and trace premix	2.90	2.90	2.90	2.90
Vitamin C (Stay C 30%)	2.80	2.80	2.80	2.80
Oxicap E2 (antioxidant)	0.20	0.20	0.20	0.20
**Nutritional composition ***				
Protein (%)	47.14	47.08	47.22	47.13
Lipid (%)	13.11	13.15	13.32	13.20
Digestible energy (MJ/kg)	16.90	18.00	16.90	16.90

Note: ^†^ Fishmeal (Anchovy) (% dry matter): crude protein 64.0%, crude lipid 10.76%, moisture 6.12% and ash 19.12%. ^‡^ Partially defatted black soldier fry larvae (% dry matter): crude protein 43.95%, crude lipid 17.23%, moisture 12.81%, and ash 8.91%. DM: dry matter. * Data were obtained from MixFeed software database program (https://softimbra.com/home, accessed on 25 April 2021).

**Table 2 biology-10-00505-t002:** Fatty acid composition (g/100 g of sample on DM-basis) of black soldier fry (BSF), partially defatted black soldier fry (PDBSF), and fishmeal (FM) used in the study.

	BSF	PDBSF	FM
C12:0	53.81	47.29	0.09
C14:0	8.87	8.55	2.91
C14:1n-5	0.15	0.16	-
C16:0	20.47	14.05	18.67
C16:1n-7	6.21	4.41	5.43
C18:0	4.65	3.10	4.35
C18:1n-9	23.37	10.58	26.10
C18:2n-6	0.13	-	0.19
C18:3n-3	5.17	0.97	3.28
C18:3n-6	0.15	0.09	0.11
C20:1n-9	0.27	0.31	0.41
C20:2	0.12	-	0.17
C20:3n-6	0.23	-	0.14
C20:4n-6	1.31	0.21	1.29
C20:3n-3	0.11	-	0.09
C20:5n-3 EPA	3.67	0.20	4.45
C22:2n-6	0.13	0.10	1.06
C22:6n-3 DHA	0.78	0.13	6.71
∑SFA	87.8	72.99	26.02
∑MUFA	30.0	15.46	31.94
∑PUFA	11.80	1.70	17.49
∑n-3	9.73	1.30	14.53
∑n-6	1.95	0.40	2.79
∑n-3/∑n-6	4.99	3.25	5.21

Note: SFA, saturated fatty acids; MUFA, monounsaturated fatty acids; PUFA, polyunsaturated fatty acids; EPA, eicosahexanoic acid; DHA, decosahexanoic acid; -, not detected.

**Table 3 biology-10-00505-t003:** List of heat shock protein and immune-relevant gene primers for qPCR analysis.

Gene	Primer Sequence (5′-3′)	References
HSP70	F: AAG GCA GAG GAT GAT GTC R: TGC AGT CTG GTT CTT GTC	[27]
HSP90	F: ACC TCC CTC ACA GAA TACC R: CTC TTG CCA TCA AAC TCC	[27]
IL-1β	F: ACAACGTCATGGAGCTCTGG R: TCTTTGTCCTTCACCGCCTC	[28]
IL-8	F: CTATTGTGGTGTTCCTGA R: TCTTCACCCAGGGAGCTTC	[29]
IL-10	F: CAGTGCAGAAGAGTCGACTGCAAG R: CGCTTGAGATCCTGAAATATA	[29]
TNF-α	F: GCCATCTATCTGGGTGCAGT R: AAAGTGCAAACACCCCAAAG	[28]
β-actin	F:GAC CTC ACA GAC TAC CTC R: GCT TCT CCT TGA TGT CAC	[30]

**Table 4 biology-10-00505-t004:** Growth performance and feed utilization of barramundi fed partially defatted BSF larvae protein and oil as FM protein and oil replacement diets.

	Control	HiP	HiO	HiPO	*p*-Value
IBW (g)	1.65 ± 0.07	1.69 ± 0.14	1.88 ± 0.05	1.78 ± 0.07	0.345
FBW (g)	30.04 ± 0.88	31.45 ± 0.96	28.04 ± 1.10	31.53 ± 3.05	0.495
WG (g)	28.39 ± 0.91	29.76 ± 1.10	26.16 ± 1.06	29.75 ± 2.98	0.460
SGR (%/d)	6.91 ± 0.14	6.98 ± 0.26	6.44 ± 0.05	6.82 ± 0.14	0.175
FCR	0.80 ± 0.01	0.76 ± 0.01	0.78 ± 0.02	0.73 ± 0.02	0.113
SR (%)	95.00 ± 2.89	90.00 ± 2.89	93.33 ± 3.33	95.00 ± 2.89	0.172

Note: IBW, initial body weight; FBW, final body weight; WG, weight gain; SGR, specific growth rate; FCR, feed conversion ratio; SR, survival rate.

**Table 5 biology-10-00505-t005:** Blood and serum biochemical parameters of barramundi fed partially defatted BSF larvae protein and oil as FM protein and oil replacement diets.

	Control	HiP	HiO	HiPO	*p*-Value
AST (u/L)	25.00 ± 10.82	26.33 ± 2.52	19.00 ± 6.93	20.67 ± 1.53	0.511
GLDH (u/L)	2.93 ± 0.72	3.53 ± 0.98	1.93 ± 0.75	2.90 ± 1.21	0.289
Cholesterol (m·mol/L)	6.43 ± 0.05	7.40 ± 0.51	6.90 ± 0.87	7.17 ± 0.66	0.312
Triglyceride (m·mol/L)	0.97 ± 0.15	1.20 ± 0.10	2.07 ± 1.76	1.33 ± 0.20	0.504
Urea (m·mol/L)	2.57 ± 0.11 ^b^	2.43 ± 0.05 ^b^	3.10 ± 0.34 ^a^	2.77 ± 0.25 ^ab^	0.030
Creatine (m·mol/L)	10.33 ± 1.15	13.67 ± 1.15	10.33 ± 0.57	11.33 ± 2.08	0.068
Glucose (g/L)	0.70 ± 0.72	0.57 ± 0.30	0.60 ± 0.62	1.10 ± 0.20	0.581
Total protein (g/L)	34.33 ± 0.57	38.67 ± 2.30	37.33 ± 4.93	36.67 ± 3.21	0.450
Albumin (g/L)	9.00 ± 0.00	10.67 ± 0.57	9.67 ± 2.08	9.67 ± 1.52	0.523
Globulin (g/L)	25.33 ± 0.57	28.00 ± 1.73	27.67 ± 2.88	27.00 ± 2.00	0.412
A/G ratio	0.36 ± 0.00	0.38 ± 0.00	0.35 ± 0.04	0.36 ± 0.04	0.670

Note: AST, Aspartate aminotransferase; GLDH, glutamate dehydrogenase; A/G, albumin/globulin. Different superscript lowercase letters within the same row denote significant differences between test diets to control (one-way ANOVA, followed by Duncan’s multiple comparison test, *p* < 0.05).

**Table 6 biology-10-00505-t006:** Fatty acid composition (mg/100 g of sample on DM-basis) of the muscle of barramundi fed partially defatted BSF larvae protein and oil as FM protein and oil replacement diets.

	Control	HiP	HiO	HiPO	*p*-Value
C10:0	0.00 ± 0.00 ^b^	4.31 ± 0.55 ^a^	1.37 ± 0.44 ^b^	4.80 ± 0.30 ^b^	0.000
C12:0	9.93 ± 1.89 ^b^	291.84 ± 34.00 ^a^	92.92 ± 25.34 ^a^	325.80 ± 17.78 ^a^	0.000
C13:0	1.05 ± 0.19	1.11 ± 0.12	1.03 ± 0.25	0.96 ± 0.04	0.939
C14:0	121.34 ± 19.61	204.98 ± 20.39	133.98 ± 29.25	188.58 ± 6.46	0.054
C14:1n-5	1.13 ± 0.22 ^b^	3.73 ± 0.36 ^a^	1.94 ± 0.51 ^b^	3.55 ± 0.15 ^a^	0.002
C15:0	26.99 ± 4.07	25.70 ± 1.95	27.55 ± 4.81	23.82 ± 0.61	0.859
C15:1	3.08 ± 0.54 ^b^	9.66 ± 0.54 ^a^	6.04 ± 1.20 ^b^	9.64 ± 0.30 ^a^	0.001
C16:0	929.03 ± 126.65	836.16 ± 66.58	881.02 ± 129.76	786.54 ± 13.89	0.759
C16:1n-7	146.38 ± 25.07	206.45 ± 20.13	153.22 ± 32.56	188.84 ± 7.52	0.280
C17:0	71.39 ± 11.52	60.84 ± 3.82	69.10 ± 7.85	59.43 ± 1.12	0.593
C17:1	7.17 ± 0.87	14.41 ± 3.92	10.85 ± 1.31	12.81 ± 2.20	0.239
C18:0	316.01 ± 37.94	290.07 ± 11.29	307.84 ± 28.97	286.10 ± 4.12	0.800
C18:1cis+trans	2042.15 ± 345.22	1025.14 ± 90.18	1882.88 ± 371.87	1223.34 ± 42.44	0.063
C18:2 trans 9	6.41 ± 3.99	4.47 ± 0.30	3.19 ± 0.11	4.60 ± 0.12	0.733
C18:2n-6	752.46 ± 116.77	413.30 ± 33.60	737.07 ± 134.66	527.64 ± 19.57	0.808
C18:3n-6	25.32 ± 4.18	22.15 ± 1.32	27.57 ± 4.66	24.86 ± 0.96	0.709
C18:3n-3	204.95 ± 36.81 ^a^	72.77 ± 6.44 ^b^	187.29 ± 38.49 ^ab^	104.47 ± 4.73 ^ab^	0.023
C18:4n-3	45.35 ± 8.30	45.64 ± 3.81	36.97 ± 7.91	29.05 ± 1.40	0.246
C20:0	20.00 ± 2.94 ^a^	9.05 ± 0.48 ^b^	16.46 ± 2.48 ^ab^	9.75 ± 0.58 ^b^	0.011
C20:1	131.24 ± 21.65 ^a^	95.25 ± 7.79 ^ab^	96.29 ± 18.27 ^ab^	58.61 ± 1.51 ^b^	0.050
C20:2	11.87 ± 1.61	9.77 ± 0.66	11.92 ± 1.68	9.71 ± 0.14	0.417
C21:0	3.85 ± 0.78	3.51 ± 0.28	3.31 ± 0.68	3.39 ± 0.78	0.893
C20:3n-6	25.28 ± 4.21	25.41 ± 1.34	25.80 ± 2.98	27.88 ± 0.72	0.891
C20:4n-6	79.21 ± 3.03 ^b^	98.07 ± 2.03 ^a^	85.14 ± 3.20 ^b^	96.87 ± 1.35 ^a^	0.002
C20:3n-3	5.11 ± 0.65	4.42 ± 0.25	5.46 ± 0.96	4.46 ± 0.09	0.562
C20:5n-3 (EPA)	128.50 ± 15.35	175.12 ± 8.43	122.07 ± 15.65	141.54 ± 2.57	0.052
C22:1n-9	10.95 ± 1.94	8.29 ± 0.76	8.59 ± 1.93	5.17 ± 0.11	0.109
C22:2	3.16 ± 3.16	4.07 ± 4.07	3.13 ± 3.13	0.00 ± 0.00	0.791
C23:0	12.43 ± 1.65	7.72 ± 3.73	12.37 ± 2.04	14.25 ± 3.91	0.499
C22:4n-6	11.49 ± 10.16	11.28 ± 10.08	12.20 ± 10.98	33.26 ± 0.57	0.072
C22:5n-3	48.61 ± 6.87	57.92 ± 3.00	46.53 ± 6.19	50.98 ± 1.38	0.433
C24:1	750.86 ± 74.80	753.86 ± 21.32	732.92 ± 67.90	652.41 ± 13.62	0.510
C22:6n-3 (DHA)	16.38 ± 2.60	10.99 ± 0.94	14.20 ± 2.25	8.98 ± 0.25	0.075

Note: EPA, eicosapentaenoic acid; DHA, docosahexaenoic acid. Different superscript lowercase letters within the same row denote significant differences between test diets to control (one-way ANOVA, followed by Duncan’s multiple comparison test, *p* < 0.05).

## Data Availability

The data presented in this study are available on request from the corresponding author.

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
