# Peer review of "Black Soldier Fly, Hermetia illucens as an Alternative to Fishmeal Protein and Fish Oil: Impact on Growth, Immune Response, Mucosal Barrier Status, and Flesh Quality of Juvenile Barramundi, Lates calcarifer (Bloch, 1790)"

_biology, 2021, doi:10.3390/biology10060505_

Round 1

Reviewer 1 Report

This manuscript investigated effect of BSF meal and oil on  growth and physiological response of barramundi. 

  1.   Check diet formulation and proximate composition.

Footnote of table 1 shows lipid content of DBSF meal was 27.23%.  Lipid content of DBSF seems to be high although BSF was defatted .

 Calculated lipid content of control and HiO are lower than values shown Table 1 as nutrient content, and much different compared to HiP and HiPO.  

2. Sum of amino acids  shown in Table 2 is considerably lower than 100 although unit is g/100g protein.  Author should explain why amino acid composition different between BSF and DBSF, although amino acid composition was expressed as g/100g protein.  Amino acid composition should be checked or reanalyzed.

 3. Fatty acid composition of BSF, DBSF, and FM.

Sum of PUFA seems to be wrong.  

Linoleic acid 18:2n-6 should be shown. 

Unit of FA is mg /100g lipid ? or mg/100g sample?

Check value of 22:4n-6 of HiP, since it is much different from other group.  If it was wrong, check n3/n6 ratio. 

4. Water quality parameter, in particular water temperature,  should be shown.

Reviewer 2 Report

The manuscript entitled “Black soldier fly, Hermetia illucens as an alternative to fishmeal protein and fish oil: effects on non-specific immunity, transcriptomic response, mucosal barrier function, and flesh quality of juvenile barramundi, Lates calcarifer (Bloch, 1790)” is in general an interesting and well-structured manuscript.

The aim of this study has been to investigate the response of fish meal and fish oil partial replacement with defatted Hermetia illucens (BSF)  insect protein and oil on several parameters including growth performance, immune response, gut and skin barrier functions and flesh quality in juvenile barramundi.

This was comprehended by conducting a nutritional trial on juvenile barramundi by assessing the effects of three experimental diets (a 30% defatted BSF larval protein, HiP, a 30% of FO displaced with BSF larvae oil, HiO, and a third one with the combination of 30% protein and 30% FO displaced with BSF larvae meal, HiPO compared to control FM/FO diet. The assessment was performed by analyzing different growth and physiological parameters including bactericidal and lysozyme activity, biochemical and molecular indicators, such as stress- and immune- relevant genes, as well as the number of the mucin cells in the gut and skin of fish fed the different diets. In general, the introduction provided is well structured focusing on the working hypotheses. The material and methods are exhaustively documented, and the experimentation is well-designed.

There is a comment related to the title, which is long and complicated in its current form. Authors are asked to present the title in a simple and clear manner. The “transcriptomic response” may mislead the readers.

My intention, if any, is to improve the current version of the discussion of the manuscript. The authors are a bit laconic; they can discuss further their results, especially on the physiological parameters examined in their manuscript and compare those to the findings of other published papers. For instance, the HSP and immune related gene expression levels are limited compared to the results found in other papers even in other fish species after fish meal and/or oil replacement. I strongly recommend the authors to compare and discuss further their results, even though to compare their results on gene expression with results on protein levels.

Additional comments:

  • Please indicate in Material and Methods the duration of the experimental trial on barramundi.
  • How do the authors decide to use b-actin as the most suitable internal control to normalize gene expression?
  • Iine 228: Replace “6” and “2” with “six” and “two”
  • Line 649: Delete (,
  • References: All scientific names must be italicized.
  • Please check the number of the lines, which appear on Table 2 and on Figure 1.

Round 2

Reviewer 1 Report

Proximate composition of diets should be analyzed.  Author mentioned that nutrient composition calculated by database program.  But, in my calculation using composition shown in footnote, lipid content of Cont and HiO diets was 2-3% lower than HiP and HiPO diets.  DBSF meal is rich in fat compared to FM.  When FM protein is replaced by DBSF meal, lipid content must increase.  However, supplemented oil to Cont is few compared to other groups.  Furthermore, why was oil levels different between Cont and  HiO?  Cont included only 27.9g/kg fish oil, but HiO contained 27.4g/kg fish oil and 10.5g/kg BSF oil.  These two groups have similar formulation, except for oil.   If author examine effect of replacement of fish oil by BFS oil, additional oil level must be same.

Author should analyze proximate composition of diets, and discuss effect of different lipid content on results of this study. 

Author Response

Response: Dear Sir, I understood your point of view. There was an obvious mistake - the fish oil content in Control diet should be 37.90 instead of 27.90. I think now the difference will be minimised a bit. However, nutritional composition estimated by database program is not always turn up with the actual composition in the final product and in that case 1-2% difference sometime we consider it insignificant. Even in statistic we hypothesis up to 5% is considerable. So, this variation should be considerable in my point of view. However, I am still learning from such types of learning criticisms.

The experiment has conducted more than a year before and further analysis with these samples may provide incorrect results.

Thank you.

Round 3

Reviewer 1 Report

Table 1; Sum of ingredients of Cont diet exceeds 1000.  Author should carefully revise. 

Author mentioned  in rev2 that unit of fatty acids is mg/100g sample.  However, sum of fatty acids shown in Table 2 is only 70-120mg/100g, which is much less compared to lipid content of FM and DBSF meal (10-17% lipid, i.e., 10000-17000mg lipid /100g sample).  Check. 
